# Scheimpflug Corneal Densitometry Patterns at the Graft–Host Interface in DMEK and DSAEK: A 12-Month Longitudinal Comparative Study

**DOI:** 10.3390/jcm12227133

**Published:** 2023-11-16

**Authors:** Antonio Ballesteros-Sánchez, Jorge Peraza-Nieves, Anna Casablanca-Piñera, Marina Rodríguez-Calvo-De-Mora, Saray Catalán-Coronado, Josep Torras-Sanvicens, Davide Borroni, José-María Sánchez-González, Carlos Rocha-De-Lossada

**Affiliations:** 1Department of Physics of Condensed Matter, Optics Area, University of Seville, 41004 Seville, Spain; 2Department of Ophthalmology, Novovision Ophthalmologic Clinic, 30008 Murcia, Spain; 3Anterior Segment Department, Institut Clinic D’Oftalmologia, Hospital Clinic Barcelona, 08036 Barcelona, Spain; peraza@clinic.cat (J.P.-N.); casablancanna@gmail.com (A.C.-P.); catalansaray13@gmail.com (S.C.-C.); jts29206@gmail.com (J.T.-S.); 4Department of Ophthalmology, Castellar Clinic, 08017 Barcelona, Spain; 5Qvision, Ophthalmology Department, VITHAS Almeria Hospital, 04120 Almeria, Spain; marocalmo@gmail.com (M.R.-C.-D.-M.); carlosrochadelossada5@gmail.com (C.R.-D.-L.); 6Ophthalmology Department, VITHAS Malaga, 29016 Malaga, Spain; 7Department of Ophthalmology, Hospital Regional Universitario, 29011 Malaga, Spain; 8School of Medicine and Health Sciences, University of Barcelona, 08036 Barcelona, Spain; 9Department of Doctoral Studies, Riga Stradins University, LV-1007 Riga, Latvia; info.borroni@gmail.com; 10Advalia Vision, Cornea Research Unit, 20145 Milan, Italy; 11Department of Surgery, Ophthalmology Area, University of Seville, 41009 Seville, Spain

**Keywords:** corneal densitometry, graft–host interface, Descemet Membrane Endothelial Keratoplasty (DMEK), Descemet Stripping Automated Endothelial Keratoplasty (DSAEK)

## Abstract

Background: To compare corneal densitometry (CD) patterns at the graft–host interface between Descemet Membrane Endothelial Keratoplasty (DMEK) and Descemet Stripping Automated Endothelial Keratoplasty (DSAEK). Corneal densitometry is a quantitative assessment that objectively evaluates corneal clarity and optical quality by measuring the light backscatter from the cornea. Methods: Fifty-one eyes that received DMEK or DSAEK surgery for corneal endothelium dysfunction were evaluated. The primary endpoint included CD patterns at the graft–host interface, which were assessed by the Pentacam HR device at the center point of the corneal horizontal meridian (CD_central_), and at six points on the central circumference of the cornea (with a total diameter of 4 mm) (CD_I,II,III,IV,V,VI_). Secondary endpoints included the best-corrected distance visual acuity (BCDVA), central corneal thickness (CCT), and graft thickness (GT). All of the evaluations were performed at follow-up appointments one, three, six and twelve months after the procedure. Results: DMEK showed a significant overall CD reduction of −7.9 ± 8.5 grayscale unit (GSU) compared to DSAEK (*p* < 0.001). In addition, the DMEK group showed significantly lower CD_Central,I,II,III,IV,V,VI_ values at follow-up appointments one, three, six and twelve months after the procedure compared to the DSAEK group (*p* < 0.001). BCDVA, CCT and GT were in favor of the DMEK group with a mean value of 0.39 ± 0.35 LogMar, 552.2 ± 71.1 µm and 11.03 ± 1.4 µm, respectively (*p* < 0.001). Conclusions: CD patterns at the graft–host interface seem to be different depending on the endothelial keratoplasty procedure. This provides specific insight into CD changes in this critical region of surgery, which may provide a better understanding of the postoperative evolution of these patients.

## 1. Introduction

Endothelial keratoplasty (EK) is currently considered the main surgical therapy of corneal endothelial dysfunction, surpassing the outcomes of penetrating keratoplasty (PK) [1,2]. EK primarily involves two surgical techniques, Descemet membrane endothelial keratoplasty (DMEK) and Descemet stripping automated endothelial keratoplasty (DSAEK), which are differentiated by the thickness of the donor tissue [3]. In DMEK, the diseased endothelial cells and the Descemet membrane are replaced by the corresponding healthy tissue, while DSAEK adds a variable portion of a donor stroma [4,5]. Consequently, DMEK seems to achieve a quicker recovery in terms of visual acuity (VA) and higher graft survival rates compared to DSAEK [6,7,8,9,10]. However, Dunbar et al. [11] showed that patient-reported vision-related quality of life was similar among DMEK and DSAEK participants. In addition, similar visual quality results were reported by Dunker et al. [12] and Torras-Sanvicens et al. [13] when comparing DMEK with ultrathin DSAEK (UT-DSAEK).

Corneal densitometry, serving as an objective metric, quantitatively gauges the transparency of the cornea, which is indicative of its optical quality [14]. This parameter measures the amount of backscattered light from the cornea, utilizing a standardized scale to provide a precise assessment of corneal clarity. The quantifiable nature of corneal densitometry allows for the comparison of corneal transparency across different clinical conditions and postoperative outcomes, facilitating a more refined analysis of corneal health and recovery processes.

However, the quality of vision is not sufficiently measured by VA and depends on other factors, such as corneal thickness, regularity, and transparency [15,16,17]. Backscattering is a well-known factor that can influence visual quality in patients with corneal diseases [18,19]. Therefore, new devices to accurately measure corneal light backscatter have emerged [20]. The corneal Scheimpflug tomography Pentacam HR device (Oculus Optikgeräte GmbH, Wetzlar, Germany) objectively measures corneal light backscatter by acquiring cross-sectional images of the anterior segment obtained through a rotating Scheimpflug camera [21]. These images are processed to generate varied grayscale maps of the corneal light backscatter, defined as corneal densitometry (CD), whose normative values have been reported recently under a comprehensive protocol [22]. Several studies have reported changes in CD patterns after EK [23,24,25,26,27]. Peraza-Nieves et al. [27] have recently compared the CD values of DMEK and DSAEK, reporting no statistically significant differences by layers or at center-paracentral concentric zones. Similar results were reported by Hirabayashi et al. [26] with no significant CD differences measured at a central 6.0 mm zone. In addition, Droutsas et al. [25] also reported that there were no significant CD differences measured at a central 6.0 mm zone after twelve months. However, the main limitation of these studies is that CD patterns in DMEK and DSAEK were evaluated with the Pentacam HR device, which performs CD measurement in the posterior layer set at 60 μm. Consequently, this layer would include the graft–host interface in DMEK, but not in DSAEK due to the greater thickness of the DSAEK graft. Therefore, it is difficult to use the Pentacam HR device to automatically compare CD changes in the posterior layers of the DMEK and DSAEK grafts and, in addition, at the graft–host interface, unless these measurements are performed manually.

The aim of this study is to compare changes in CD patterns, specifically at the graft–host interface in DSAEK and DMEK, in addition to correlating these changes with the best-corrected distance visual acuity (BCDVA), central corneal thickness (CCT) and graft thickness (GT).

## 2. Materials and Methods

### 2.1. Study Design and Patients

This prospective, longitudinal and comparative study was carried out at the Barcelona Hospital Clinic from January 2017 to December 2019. It was approved by the institutional review board (ID: HCB/2021/1197) and adhered to the tenets of the Declaration of Helsinki. Before initiating the study, informed consent was obtained from each patient. All of the patients diagnosed with corneal endothelium dysfunction requiring EK at Barcelona Hospital Clinic were included in the study. Only cases with sufficient clinical follow-up data at 1, 3, 6 and 12 months were analyzed. The exclusion criteria included previous corneal surgeries, in addition to complex pathologies of the anterior or posterior segment, such as uncontrolled glaucoma, active vascular retinal diseases, and manifest uveitis, among others. Furthermore, eyes showing complications after EK, such as primary graft failure, immune reaction, graft rejection, and necessity for re-grafting during the course were also excluded.

### 2.2. Surgical Procedures

Regarding DSAEK surgery, a precut corneal graft (Eye Bank, Barcelona, Spain) less than 150 μm thickness, stored in a specific organ culture medium (Eurobio Scientific, Les Ulis, France) was obtained. The precut corneal graft was trephined at 8.0–8.5 mm diameter with a Barron Corneal Punch (Katena Products, NJ, USA) and placed into a busin glide spatula (Moria SA, Massy, France). In addition, the corneal graft was marked on the stromal side with the letter “S” or “F” according to the surgeon’s preference. The surgical procedure followed the technique described by Busin et al. [28] At the end of the surgery, the incisions were sutured with Nylon 10-0 and either air or sulfur hexafluoride 20% (SF6) tamponade was injected into the anterior chamber according to the surgeon’s preference.

In DMEK surgery, the corneal graft was pre-stripped at the Barcelona Eye Bank and sent to the Barcelona Hospital Clinic in a specific organ culture medium (Eurobio Scientific, Les Ulis, France). In the theater room, the graft was trephined at 8.0–8.5 mm diameter with a Barron Corneal Punch (Katena Products, NJ, USA) and then separated from the stromal bed with Troutman forceps (Moria SA, Massy, France). The DMEK procedures were performed under retrobulbar anesthetic following the “no-touch” Melles’s technique [29]. However, tamponade was made with either air or SF6 depending on the choice of each surgeon. The surgical procedures were performed by two expert corneal surgeons (JPN and JTS).

### 2.3. Measurement

The CD patterns were obtained using the Pentacam HR device (Oculus Optikgeräte GmbH). This device emits a blue light beam (475 nm) as an integrated camera captures 25,000 points simultaneously and generates a series of 25 images (1003 × 520 pixels) of various corneal meridians [21]. The software automatically identifies the corneal apex and measures the CD, which is expressed in a grayscale unit (GSU) that ranges from 0 (minimum light backscattering) to 100 (maximum light backscattering) [22]. In addition, Pentacam HR also allows for manual measurements of the CD patterns at any point on the cornea. To acquire the CD on the DMEK graft–host interface, the measurement plane was positioned approximately 15 ± 5 μm anterior to the cornea-aqueous humor interface, as shown in Figure 1A. The thickness of Descemet’s membrane and endothelium in humans is believed to be apporiximately 10 μm and 5 μm in adult corneas [30,31], respectively. Due to limitations in the imaging software, which does not allow for precise micrometer-by-micrometer adjustments throughout the corneal thickness, the measurement process was initiated with the mouse cursor placed at the cornea-aqueous humor interface. Afterwards, the mouse cursor was incrementally moved anteriorly in steps of several microns until it reached the closest plane, approximately 15 ± 5 μm anterior to the cornea-aqueous humor interface. Regarding DSAEK, the graft–host interface was visually localized with a gray-white color change in the image, as shown in Figure 1B.

In the present study, CD patterns at the graft–host interface were evaluated in the following locations: (1) at the central point of the corneal horizontal meridian; and (2) at 6 points on the central circumference of the cornea with a total diameter of 4 mm (we focused on the central 4 mm because, in line with established literature, this is critical for visual quality and typically presents fewer post-surgical complications) [32], which were defined as “CD_Central_” and “CD_I,II,III,IV,V,VI_”, respectively, as shown in Figure 1. In addition, BCDVA, CCT, and GT were also evaluated. All of the variables were examined at 1, 3, 6 and 12 months after surgery under standard light conditions in the same room and by the same trained optometrist.

### 2.4. Statistical Analysis

Statistical analyses were performed with SPSS statistics software, version 28.0 (IBM Corporation, New York, NY, USA). A sample size of 70 patients was estimated using the GRANMO calculator, version 7.12 (Municipal Institute of Medical Research, Barcelona, Spain). The estimation was based on a statistically significant paired difference at 95% confidence with 80% power of 3.35 ± 4.72 GSU in CD, as in previous studies. Continuous variables were displayed as the mean ± standard deviation (SD) with interquartile ranges [IQRs], whereas ordinal categorical variables were expressed as frequencies (n) and percentages (%). After testing for normality and homogeneity of variance, a repeated-measures ANOVA (parametric) or a Friedman test (non-parametric) was performed to compare intra-group CD. Within each group, the increment (Δ) in CD also was calculated. It was defined as the changes from the 1-month (1-mo) to the 12-month (12-mos) follow-up, or ΔCD = 12-mos − 1-mo. The inter-group CD was analyzed with the unpaired Student *t*-test (parametric) or the Mann-Whitney U test (non-parametric). Between each group, the differences in corneal densitometry were calculated as ΔCD_DMEK group_ − ΔCD_DSAEK group_. The Pearson (parametric) or the Spearman Rho correlation coefficient (non-parametric) was used to analyze the correlations between variables. In addition, stepwise multiple linear regression analysis was performed to detect the influential factors in ΔCD. The level of significance was *p* < 0.05 for all comparisons.

## 3. Results

Seventy-three eyes of 73 patients, 28 (50.9%) men and 45 (49.9%) women, with a mean age of 72.9 ± 9.4 (46–96) years, were enrolled in the study. Fuchs endothelial dystrophy (41.6%) and pseudophakic bullous keratopathy (31.4%) were the main causes of EK. The graft size was 7.44 ± 1.56 (7.5–8.5) mm and 8.21 ± 0.48 (7.25–8.5) mm for DSAEK and DMEK, respectively. No significant differences in demographic characteristics, ocular diseases and graft size were detected between the groups. However, 22 (30.1%) patients did not complete the study due to incomplete follow-up data. The characteristics of patients who completed the study are presented in Table 1. After the 12-month follow-up, the patients in our study who underwent DMEK demonstrated a significantly enhanced BCDVA when juxtaposed with those treated with DSAEK. This superiority in visual outcome is attributed to the thinner graft profile of DMEK procedures, as reflected in the markedly reduced CCT and GT, with statistical significance (*p* < 0.001) evident in all comparative measures.

### Corneal Densitometry Patterns

Intra-group CD_Central_ is presented in Figure 2. Regarding the DMEK group, statistically significant differences were observed between CD_Central_ when measured at the 1 and 12 month follow-ups (*p* = 0.012). In DMEK surgery, only endothelium is transplanted (i.e., without the stroma). This allows DMEK tissue to increase the cornea’s clarity over time. In DSAEK, the presence of the stroma reduces the possibility of a decrease in CD. Additionally, DSAEK influences the refractive outcome, whereas typically DMEK does not. Upon thorough analysis, our results indicated that there were no statistically significant differences in the changes in central corneal densitometry (ΔCD) across zones I through VI when comparing the DMEK and DSAEK groups. A comparison between the CD patterns of both groups is presented in Table 2. When integrating all the CD measurements performed at the different follow-up times into a single variable, the DMEK group showed a significant CD reduction of −7.9 ± 8.5 GSU compared to the DSAEK group (*p* < 0.001). In addition, patients undergoing DMEK showed significantly lower CD_central,I,II,III,IV,V,VI_ values than those undergoing DSAEK at the 1, 3, 6 and 12 month follow-ups (*p* < 0.01 for all comparisons). However, no statistically significant differences between the groups were found in the ΔCD_central,I,II,III,IV,V,VI_.

Regarding single correlations in the DSAEK group, ΔCCT displayed a significant correlation to ΔCD_Central_ (r = 0.51, *p* = 0.026), while ΔGT correlated significantly to ΔCD_II_ (r = 0.45, *p* = 0.05) and ΔCD_IV_ (r = 0.58, *p* = 0.009). However, no significant correlations were found in the DMEK group. Regarding multiple correlations in the DSAEK group, ΔCCT had the strongest association with ΔCD_Central_ (R^2^ = 0.22, 95% CI 0.010 to 0.14, *p* = 0.026), while ΔGT was associated with ΔCD_II_ (R^2^ = 0.16, 95% CI 0.00 to 0.33, *p* = 0.05) and ΔCD_IV_ (R^2^ = 0.31, 95% CI 0.04 to 0.21, *p* = 0.05). In the DMEK group, ΔCCT was also associated with ΔCD_Central_ (R^2^ = 0.18, 95% CI 0.02 to 0.12, *p* = 0.009), as well as with ΔCD_III_ (R^2^ = 0.16, 95% CI 0.01 to 0.06, *p* = 0.015), ΔCD_IV_ (R^2^ = 0.16, 95% CI 0.01 to 0.04, *p* = 0.013) and ΔCD_V_ (R^2^ = 0.16, 95% CI 0.01 to 0.04, *p* = 0.015).

In the DSAEK group, our findings highlighted a moderate correlation between the ΔCCT and ΔCD_Central_ (R_2_ = 0.22), suggesting a relationship between postoperative corneal thickness and transparency. Additionally, ΔGT correlated notably with densitometry in peripheral zones II and IV, reflecting the influence of graft structure on corneal clarity. Similarly, the DMEK group exhibited a consistent correlation between ΔCCT and ΔCD_Central_ (R_2_ = 0.18), with analogous correlations in zones III, IV, and V. These associations indicate a pattern in corneal responses to surgical interventions, with regional variations in clarity linked to changes in corneal and graft thickness post-DMEK. These insights help clarify how corneal structural changes post-surgery can influence optical quality.

## 4. Discussion

EK has reached new frontiers for the visual rehabilitation of patients with corneal endothelial dysfunction compared to traditional PK [4,33,34]. DMEK provides a complete anatomical restoration of the cornea, whereas DSAEK is an additive procedure in which a variable amount of stroma from the donor is transplanted. The collagen fibrils in a healthy cornea shows an exquisite disposition that enables its transparency [35]. In DSAEK, the donor stroma is attached to the posterior stroma of the host, creating a stroma-stroma graft–host interface that may induce an increase in corneal light backscattering. This stroma-stroma interface may potentially influence the visual outcome of the procedure and could ultimately replace the DSAEK graft [18,19,36]. The analysis of the graft–host interface has attracted increasing interest over the last few years [37,38]. The Pentacam HR device automatically locates the corneal apex and measures corneal light backscattering using CD maps, which analyze four corneal concentric zones (central zone: 2 mm; first ring: 2–6 mm; second ring: 6–10 mm; and third ring: 10–12 mm) at different depths (anterior: 120 μm; central: 120–60 μm; and posterior: 60 μm) [21]. Several studies have used this device to analyze the changes in CD patterns between DMEK and DSAEK [23,24,25,26,27]. However, Pentacam HR is unable to automatically analyze corneal light backscattering at the graft–host interface, which represents the region undergoing the most significant changes during the postoperative period [39]. Consequently, the aim of this study is to compare CD patterns at the graft–host interface between DMEK and DSAEK over a period of 12 months.

In this study, CD patterns only improved significantly in the DMEK group after 12 months of follow-ups. In addition, the DMEK group showed significantly lower CD values compared to the DSAEK group at the 1, 3, 6 and 12 month follow-ups. Schaub et al. [24] reported similar results after 2 years of follow-ups, which suggest the relevance of the long-term follow-up of patients who have undergone EK. However, our study also reported that ΔCD was not significantly different between the two groups. Similar results were reported by Hirabayashi et al. [26] who showed that there was no significant difference in the improvement of CD between DMEK and UT-DSAEK after 12 months of follow-ups. Despite this, lower CD values in the DMEK group could explain a better BCDVA compared to the DSAEK group, as reported in this study. In addition, several studies have reported a significant, positive correlation between CD and BCDVA in patients with EK [24,40,41]. However, our study found no significant correlation between either of the variables. Therefore, there may be other factors involved which explain the superior BCDVA of DMEK over DSAEK, such as high-order aberrations (HOAs) [36,42,43], CCT [44,45], and GT [46].

Regarding HOAs, Hayashi et al. [42] and Duggan et al. [43] reported that the DMEK group had significantly fewer HOAs of the posterior corneal surface compared to the DSAEK and UT-DSAEK groups, which significantly correlated with a better BCDVA. Similar results were reported by Dirisamer et al. [36] who found that coma and trefoil at the posterior corneal surface were lower in DMEK than in DSAEK. Concerning CCT, Machalińska et al. [44] reported that central thickness positively correlated with BCDVA, and CCT values in the DMEK group were significantly lower than in the DSAEK group. In addition, a recent study by Moskwa et al. [45] proposed that a CCT of ≥ 625 μm was associated with worse BCDVA following DMEK. Pertaining to GT, Tourabaly et al. [46] evaluated the influence of GT on BCDVA after different EK techniques, reporting that a GT of 16 ± 5 μm (DMEK group) achieves significantly better BCDVA compared to a GT of 78 ± 14 μm (UT-DSAEK group) and 162 ± 9 μm (DSAEK group), respectively. It is important to mention that the results reported by Machalińska et al. [44] and Tourabaly et al. [46] are in line with the CCT and GT values reported in our study, which may also explain the BCDVA results in the DMEK and DSAEK groups. In addition, stepwise multiple linear regression analysis showed that ΔCCT and ΔGT had an association with ΔCD in both groups. However, they only could predict transparency of the graft–host interface in 19.3 ± 5.6%, which suggest that other factors may contribute significantly to the variation in transparency of the graft–host interface, such as debris, scarring [47] or Descemet membrane remnants in the interface [36], irregularities in the DSAEK graft, and the stroma-stroma interface in DSAEK [36]. Therefore, future research is needed to identify and understand these additional factors which may affect CD in patients undergoing EK.

In the postoperative period, corneal densitometry readings may be influenced by a range of inflammatory and fibrotic factors, including edema, corneal stromal fiber hyperplasia, and inflammation. These parameters, integral to the cornea’s healing and recovery processes, can affect transparency and are essential considerations when evaluating the outcomes of DMEK and DSAEK surgeries [14]. The interpretation of densitometry must, therefore, be contextualized within the framework of the cornea’s response to surgical intervention and its subsequent repair mechanisms.

### Strengths and Limitation

This study has some limitations that need to be addressed. The relative loss of follow-ups, as well as the small number of cases in the DSAEK group may have influenced the results. To the best of our knowledge, this is the first study to analyze CD patterns at different points of the graft–host interface in DMEK and DSAEK. Comparisons with other studies are limited because they analyze CD by regions at different depths, whereas we manually performed the CD measurements aiming at the graft–host interface. We considered that, nowadays, this was the only possible way to evaluate the graft–host interface with this device. It should be noted that this study did not evaluate the relationship between DSAEK graft thickness and corneal densitometry values, thus no data on the influence of residual stromal thickness on densitometry outcomes are available in our findings. The graft–host interface in EK seems to be important with respect to visual recovery, therefore, there is a need for new devices that automatically analyze CD at different points of the graft–host interface. Although confocal microscopy offers more precision in identifying the graft–host interface, it was not utilized in this study. Future research might benefit from employing this technique to enhance the accuracy of measurements.

Overall, further studies are needed to confirm the findings reported in this study. In addition, it would be of interest to compare CD at the graft–host interface in DSAEK and DMEK with healthy a population to determine which technique achieves a better restoration of the corneal anatomy. While preoperative corneal densitometry was not assessed in this study, we acknowledge that baseline corneal clarity and high-order aberrations (HOAs) may influence postoperative densitometry patterns. Future studies could benefit from incorporating preoperative densitometry measurements to further elucidate their impact on surgical outcomes.

## 5. Conclusions

In conclusion, CD patterns evaluated at different points of the graft–host interface were significantly lower in the DMEK group compared to the DSAEK group after 12 months of follow-ups. In addition, ΔCCT and ΔGT were significant correlated with ΔCD. At the graft–host interface, corneal stromal transparency measured by corneal Densitometry is greater in DMEK than in DSAEK. Postoperative corrected visual acuity was better in DMEK than in DSAEK and seems to relate little to corneal densitometry. However, although ΔBCDVA was significantly better in the DMEK group, it was not significantly correlated with ΔCD, which suggests an impact of other factors on the final visual outcomes. Therefore, it is important to consider multiple factors when assessing CD patterns in patients undergoing EK, as well as to analyze them in follow-ups over the long-term.

## Figures and Tables

**Figure 1 jcm-12-07133-f001:**
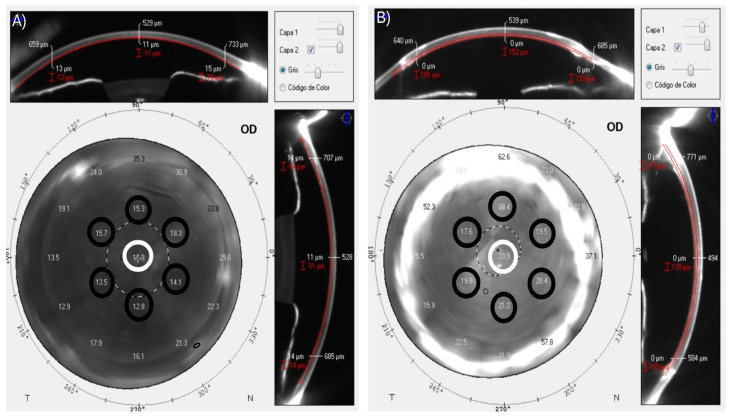
Manual measurements of CD patterns at the graft–host interface after two types of endothelial keratoplasty; (**A**) DMEK, the graft–host interface was located approximately 15 ± 5 μm anterior to the cornea-aqueous humor interface and (**B**) DSAEK, the graft–host interface was visually localized with a gray-white color change in the image. CD patterns were assessed in the following locations: (1) at the central point of the corneal horizontal meridian and (2) at 6 points on the central circumference of the cornea with a diameter of 4 mm, which were defined as “CD_Central_” (white circles) and “CD_I,II,III,IV,V,VI_” (black circles), respectively.

**Figure 2 jcm-12-07133-f002:**
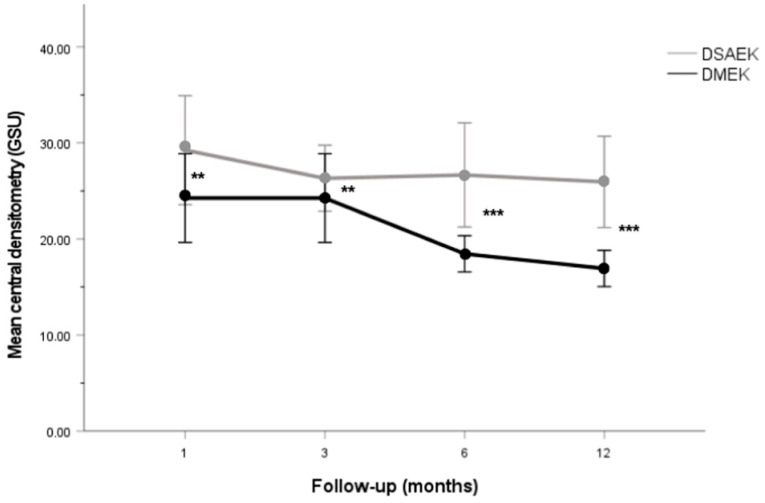
Line graph showing CD_Central_ at different follow-up times in the DSAEK and DMEK groups. CD_Central_ is significantly lower in the DMEK group compared to the DSAEK group at the 1, 3, 6 and 12 month of follow-ups. ** *p* < 0.01. *** *p* < 0.001.

**Table 1 jcm-12-07133-t001:** Study population characteristics.

Characteristics	DSAEK (n = 19)	DMEK (n = 32)	*p*
**Demographics, mean ± SD (IQR) or n (%)**			
Age (years)	69.84 ± 10 (52–88)	74.56 ± 11.75 (46–94)	0.150 ^2^
Sex, male/female	11 (57.9)/8 (42.1)	15 (46.9)/17 (53.1)	0.447 ^3^
**Surgery, mean ± SD (IQR)**			
Button size (mm)	7.66 ± 1.37 (2.2–8.5)	8.02 ± 0.37 (7.25–8.5)	0.336 ^4^
Graft Thickness (µm) ^1^	120.81 ± 20.42 (82–197)	11.03 ± 1.4 (8–19)	**<0.001** *^,4^
CCT (μm) ^1^	635.21 ± 80.02 (531–895)	552.2 ± 71.10 (429–987)	**<0.001** *^,4^
BCDVA (LogMar) ^1^	0.66 ± 0.36 (0–1.3)	0.39 ± 0.35 (0–1.3)	**<0.001** *^,4^
**Ocular disease, n (%)**			
Fuchs endothelial dystrophy	6 (31.6)	16 (50)	0.199 ^3^
Pseudophakic bullous keratopathy	8 (42.1)	8 (25)	0.203 ^3^
Descemet Membrane detachment	2 (10.5)	2 (6.3)	0.583 ^3^
Herpetic endotheliitis	0 (0)	3 (9.4)	0.169 ^3^
Corneal decompensation	3 (15.8)	3 (9.4)	0.492 ^3^

BCDVA, best corrected distance visual acuity; CCT, corneal central thickness; DMEK, Descemet membrane endothelial keratoplasty; DSAEK, Descemet stripping automated endothelial keratoplasty; GDD, glaucoma drainage device. IQR, interquartile ranges. LogMar, logarithm of the minimum angle of resolution. ^1^ Defined as the mean value during the follow-up period; ^2^ Paired student *t*-test; ^3^ Chi-square test (χ^2^); ^4^ Mann-Whitney U test; ***** Statistical significance level with a *p* value < 0.05.

**Table 2 jcm-12-07133-t002:** Corneal densitometry comparison at different points of the graft–host interface between DSAEK and DMEK.

Follow-Up	Corneal Densitometry Location ^1^							
		Central, GSU	I, GSU	II, GSU	III, GSU	IV, GSU	V, GSU	VI, GSU
1 month	**DSAEK**	29.25 ± 11.78 [16.1–71.8]	30.5 ± 6.05 [21.2–48.4]	31.38 ± 9.26 [21.4–58.8]	25.98 ± 5.59 [17.6–36.2]	24.28 ± 4.31 [19.2–35]	27.45 ± 8.43 [17.9–47]	28.89 ± 13.48 [14.6–77.4]
	**DMEK**	24.26 ± 12.8 [14.1–77.6]	18.67 ± 5.1 [11.2–36.4]	20.55 ± 7.12 [12.3–42]	18.58 ± 7.56 [11.1–44.7]	17.77 ± 6 [10.3–33.3]	17.43 ± 5.43 [11.2–36.7]	19.04 ± 5.33 [12.6–38.5]
	*p* value	**0.005** *^,2^	**<0.001** *^,2^	**<0.001** *^,2^	**<0.001** *^, 3^	**<0.001** *^,2^	**<0.001** *^,2^	**<0.001** *^,2^
3 months	**DSAEK**	26.32 ± 7.12 [18.8–45.5]	23.77 ± 4.86 [14.5–31.9]	25.56 ± 7.55 [15.1–44.1]	22.31 ± 6.28 [14–35.5]	21.33 ± 4.39 [16.5–32]	24.52 ± 6.89 [16.9–44.1]	30.62 ± 14.19 [14.1–54.9]
	**DMEK**	22.22 ± 11.89 [11–65.9]	17.96 ± 6.5 [4.7–43.5]	19.32 ± 7.8 [11.6–54.5]	16.95 ± 7.53 [9.1–49.3]	16.91 ± 7.59 [10.4–50.2]	16.44 ± 4.85 [10.9–33.9]	19.04 ± 7.61 [11.7–49.7]
	*p* value	**0.002** *^,2^	**<0.001** *^,3^	**0.004** *^,2^	**0.006** *^,3^	**<0.001** *^,2^	**<0.001** *^,2^	**<0.001** *^,2^
6 months	**DSAEK**	26.67 ± 11.25 [16.9–61.2]	24.23 ± 6.32 [12.9–35.9]	25.52 ± 7.7 [18–42.7]	21.61 ± 6.58 [12.9–39.4]	20.98 ± 5.28 [15.4–34]	23.15 ± 6.89 [16.7–44]	25.59 ± 11.31 [12.4–55.6]
	**DMEK**	18.45 ± 5.23 [11.4–32.2]	16.31 ± 6.33 [4.5–35.4]	18.7 ± 6.61 [11.6–46.1]	15.49 ± 4.41 [10.2–29.5]	17.41 ± 15.18 [10.5–98]	15.02 ± 3.24 [11.6–26.6]	17.31 ± 5.98 [11.4–40.4]
	*p* value	**<0.001** *^,2^	**<0.001** *^,3^	**<0.001** *^,2^	**<0.001** *^,2^	**<0.001** *^,2^	**<0.001** *^,2^	**<0.001** *^,2^
12 months	**DSAEK**	25.94 ± 9.84 [13.7–50.2]	25.07 ± 13.75 [12.4–75.9]	25.21 ± 11.41 [14.3–57.3]	22.59 ± 8.71 [11.7–40.9]	20.08 ± 5.02 [13.6–31.7]	21.76 ± 6.35 [13.9–34]	25.89 ± 16.3 [13–80.6]
	**DMEK**	16.92 ± 5.23 [2.12–30.4]	17.24 ± 4.57 [8.6–26.7]	17.05 ± 5.06 [2.83–30.6]	15.95 ± 5.17 [10.7–37.6]	13.9 ± 3.89 [9.5–27.3]	14.9 ± 5.02 [10.2–34.2]	16.43 ± 4.11 [12.1–31.1]
	*p* value	**<0.001** *^,2^	**0.003** *^,2^	**0.004** *^,3^	**0.002** *^,3^	**<0.001** *^,3^	**<0.001** *^,2^	**<0.001** *^,2^

DMEK, Descemet membrane endothelial keratoplasty; DSAEK Descemet stripping endothelial keratoplasty; GSU, Grayscale unit. ^1^ Measured at the central point of the graft–host interface, as well as in I, II, III, IV, V, and VI points corresponding to a central circumference with a total diameter of 4 mm; ^2^ Mann-Whitney U test; ^3^ Unaired student *t*-test; * Statistical significance level with a *p* value < 0.05.

## Data Availability

The data presented in this study are available on request from the corresponding author. The data are not publicly available due to their containing information that could compromise the privacy of research participants.

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
