# Peer review of "Scheimpflug Corneal Densitometry Patterns at the Graft–Host Interface in DMEK and DSAEK: A 12-Month Longitudinal Comparative Study"

_jcm, 2023, doi:10.3390/jcm12227133_

Round 1

Reviewer 1 Report

Comments and Suggestions for Authors

*The authors aimed to compare corneal densitometry patterns at the graft-host interface between Descemet Membrane Endothelial Keratoplasty and Descemet Stripping Automated Endothelial Keratoplasty. I think some issues should be addressed before further consideration.

*I think the most considerable concern is regarding the correct segmentation of the graft area since it has been done manually at a range of 10µm, while this amount is approximately equal to a DMEK donor thickness.

*Why did you choose the central 4mm, while the size of trephination was about 8mm?

*How do you justify the lines 185-188?

*Superiority of DMEK outcomes over DSAEK is a well-known finding, however it possesses more technical challenges.

Comments on the Quality of English Language

Minor polishing is required.

Reviewer 2 Report

Comments and Suggestions for Authors

1. Title is very long to consider putting DMEK and DSAEK, something like this: Scheimpflug  Corneal Densitometry Patterns at the Graft-Host Interface in DMEK and DSAEK: A Longitudinal Comparative Study.

2. Abstract: define in one sentence that corneal densitometry is a quantifiable measure of corneal clarity.

3. Abstract: Put in the Conclusions, something like this: At the graft-host interface, corneal stromal transparency measured by corneal Densitometry is greater in DMEK than in DSAEK. Postoperative corrected visual acuity was better in DMEK vs DSAEK and seems little related to corneal densitometry.

Consider including the following reference: Sela TC, Iflah M, Muhsen K, Zahavi A. Descemet membrane endothelial keratoplasty compared with ultrathin Descemet stripping automated endothelial keratoplasty: a meta-analysis. BMJ Open Ophthalmol. 2023 Nov;8(1):e001397. doi: 10.1136/bmjophth-2023-001397. PMID: 37914389.

4. Line 37 and 192: Where it says GSU value put greyscale unit (GSU)

 5. Introduction: Define that corneal densitometry, what it really measures is corneal transparency and is a quantifiable parameter (Dedicate a paragraph of the introduction to this concept).

6. Results: Line 181-183. Write this paragraph better: reported significantly superior BCDVA compared to DSAEK patients. Due to the increased thickness of the DSAEK graft DMEK showed lower CCT and GT (P < 0.001 for all comparisons).

7. Table 1. Graft Thickness, (µm )  en vez de  (mm).

8. Results. Line 195. Consider explaining something better, it's confusing: However, no statistically significant differences were found in the central ΔCD, I, II, III, IV, V, VI between both groups.

9. Results line 200-206. Include a few lines of text that explain to the reader the result and meaning of all these correlations, more than just numbers:  found in the DMEK group. Regarding multiple correlations in the DSAEK group, ΔCCT had the strongest association with ΔCD Central (R2 = 0.22, 95% CI 0.010 to 0.14, P = 0.026), while ΔGT was associated with ΔCD II (R2 = 0.16, 95% CI 0.00 to 0.33, P = 0.05) and ΔCD IV (R2 = 0.31, 95% CI 0.04 to 0.21, P = 0.05). In the DMEK group, ΔCCT was also associated with ΔCD Central (R2 = 0.18, 95% CI 0.02 to 0.12, P = 0.009), as well as with ΔCD III (R2 = 0.16, 95% CI 0.01 to 0.06, P = 0.015), ΔCD IV (R2 = 0.16, 95% CI 0.01 to 0.04, P = 0.013) and ΔCD V (R2 = 0.16, 95% CI 0.01 to 0.04, P = 0.015).

10 . Discussion. Line 264-276. There are paragraphs that have many acronyms and few full stops. Terms such as transparency of the graft-host interface would be appreciated. coma and trefoil at the posterior corneal surface were lower in DMEK than in DSAEK. Concerning CCT, Machalińska et al. [43] reported that CCT was positively correlated with BCDVA, as well as CCT values in the DMEK group were significantly lower than in the DSAEK group. In addition, a recent study by Moskwa et al. [44] proposed that a CCT of ≥ 625 μm was associated with worse BCDVA after DMEK. Pertaining to GT, Tourabaly et al. [45] evaluated the influence of GT on BCDVA after different EK techniques, reporting that a GT of 16 ± 5 μm (DMEK group) achieves significantly better BCDVA compared to a GT of 78 ± 14 μm (UT-DSAEK group) and 162 ± 9 μm (DSAEK group), respectively. It is important to mention that the results reported by Machalińska et al. [43] and Tourabaly et al. [45] are in line with the CCT and GT values reported in our study, which may also explain the BCDVA results in the DMEK and DSAEK groups. In addition, stepwise multiple linear regression analysis showed that ΔCCT and ΔGT had an association with ΔCD in both groups. However, they could only predict ΔCD in 19.3 ± 5.6 %, which suggest that

11. Little is reflected in the Discussion. That corneal densitometry can be influenced by inflammatory and fibrosis parameters after surgery such as: edema, corneal stromal fiber hyperplasia, and corneal inflammation. And repair and recovery processes after DMEK or DSAEK surgery.

Reviewer 3 Report

Comments and Suggestions for Authors

Dear Authors , 

I read with attention the paper on Scheimpflug Corneal Densitometry patterns at the graft host interface in DMEK and DSAEK and here you'll find some comments.

- Abstract line 31 :  51 eyes involved in the study  with preoperative diagnosis of "corneal endothelial disfunction"were evaluated.... did you measure with Pentacam also the preoperative values of corneal densitometry in these eyes? As You explained in the discussion , other factors may contribute to the difference in corneal densitometry patterns , like HOA . May preoperative corneal densitometry patterns , variable according to anatomy and pathology, could influence this value?  What do you think according to the outcome of this study? 

-line 51 : involveS

- line 56: obtain 

- line 67: backscattering

- in  my opinion, the manual measurement of the graft host interface is the biggest limitation of the study, even more because the interface was calculated based on its subjective visual identification. Did you evaluate the chance to use Confocal microscopy to identify the interface more precisely? 

-lines 103-104 : DSAEK graft thickness included in the study was less than 150 microns : did you find any difference in CD values according to residual stromal thickness of the graft? 

line 213: wich instead of with

line 218:  I suggest visual outcome of the procedure instead of visual quality 

Comments on the Quality of English Language

Dear Editors, I read with attention the paper by Ballesteros Sanchez et al. and I found it of  interest for the readers . To my opinion, a  minor revision work is needed together with an english editing. 

Round 2

Reviewer 2 Report

Comments and Suggestions for Authors

The authors have made the changes suggested by the reviewer.